# Antiviral Activity of Acetylsalicylic Acid against Bunyamwera Virus in Cell Culture

**DOI:** 10.3390/v15040948

**Published:** 2023-04-11

**Authors:** Sara Yolanda Fernández-Sánchez, José P. Cerón-Carrasco, Cristina Risco, Isabel Fernández de Castro

**Affiliations:** 1Cell Structure Laboratory, Centro Nacional de Biotecnología, CSIC, Campus de Cantoblanco, 28049 Madrid, Spain; syfernandez@cnb.csic.es; 2Centro Universitario de la Defensa, Universidad Politécnica de Cartagena, C/Coronel López Peña s/n, Base Aérea de San Javier, Santiago de la Ribera, 30720 Murcia, Spain; jose.ceron@cud.upct.es

**Keywords:** bunyavirus, Bunyamwera virus, viral RNA polymerase, viral replication organelle, acetylsalicylic acid (ASA), antiviral, high-throughput screening, molecular modeling, drug repurposing, electron microscopy

## Abstract

The *Bunyavirales* order is a large group of RNA viruses that includes important pathogens for humans, animals and plants. With high-throughput screening of clinically tested compounds we have looked for potential inhibitors of the endonuclease domain of a bunyavirus RNA polymerase. From a list of fifteen top candidates, five compounds were selected and their antiviral properties studied with Bunyamwera virus (BUNV), a prototypic bunyavirus widely used for studies about the biology of this group of viruses and to test antivirals. Four compounds (silibinin A, myricetin, L-phenylalanine and p-aminohippuric acid) showed no antiviral activity in BUNV-infected Vero cells. On the contrary, acetylsalicylic acid (ASA) efficiently inhibited BUNV infection with a half maximal inhibitory concentration (IC_50_) of 2.02 mM. In cell culture supernatants, ASA reduced viral titer up to three logarithmic units. A significant dose-dependent reduction of the expression levels of Gc and N viral proteins was also measured. Immunofluorescence and confocal microscopy showed that ASA protects the Golgi complex from the characteristic BUNV-induced fragmentation in Vero cells. Electron microscopy showed that ASA inhibits the assembly of Golgi-associated BUNV spherules that are the replication organelles of bunyaviruses. As a consequence, the assembly of new viral particles is also significantly reduced. Considering its availability and low cost, the potential usability of ASA to treat bunyavirus infections deserves further investigation.

## 1. Introduction

Emerging and re-emerging viruses are a major health concern, and the validation of broad-spectrum antivirals that can be quickly applied to the clinic is, therefore, a priority. There are many viruses that can cause disease in humans. In particular, the World Health Organization’s (WHO) current list of prioritized diseases [1] includes three diseases caused by coronaviruses (SARS-CoV, MERS-CoV and SARS-CoV-2) and three more caused by members of the order *Bunyavirales*: Rift Valley Fever virus (RVFV), Crimean–Congo Hemorrhagic Fever virus (CCHFV) and Lassa Fever Virus (LFV). As a consequence of the intense research developed after the emergence of SARS-CoV-2, three antivirals have been approved to treat the associated pathology known as COVID-19: remdesivir, molnupiravir and paxlovid, the latter being a mixture of ritonavir and nirmatrelvir [2,3,4,5], and several more are currently being evaluated in clinical trials (https://clinicaltrials.gov/, accessed on 5 April 2023). However, there are no vaccines used worldwide, and no specific antivirals have been approved so far to treat bunyavirus infections. Ribavirin and Favipiravir, two inhibitors of viral RNA polymerases, are currently being studied in ongoing clinical trials for Lassa fever and Crimean–Congo hemorrhagic fever (https://clinicaltrials.gov/, accessed on 5 April 2023).

The *Bunyavirales* order comprises more than 500 viruses that are generally defined as bunyaviruses. With the exception of Hantaviruses and Arenaviruses, all viruses in the *Bunyavirales* order are transmitted by arthropods [6]. Like many other RNA viruses, bunyaviruses replicate their genome in association with intracellular membranes [7,8]. BUNV viral replication complexes (VRC) assemble in Golgi-derived single-membrane vesicles or spherules, which constitute the viral replication organelle (RO) and which are often connected with tubular structures [7,9,10]. Immature viral particles also assemble in Golgi membranes and undergo two maturation steps, the first one in the *trans*-Golgi compartment and the second one during virus egress that renders fully infectious virions [11]. All these steps are susceptible to being blocked by antiviral compounds. Antivirals are molecules that interfere with a step of the virus life cycle. There are basically two types: (i) direct acting antivirals (DAAs) that target viral components, and (ii) drugs against cell factors and pathways. DAAs are very specific and efficient for a particular virus or group of related viruses. Their main disadvantage is that virus variants might escape. On the other hand, inhibitors of cell components used by viruses should be efficient against emerging variants. However, a blockade of cell pathways can produce more severe secondary effects [12].

The incidence of bunyaviruses is growing due to several factors, such as climate change and globalization which favor spreading of arthropod vectors [13,14,15]. Thus, reports of novel bunyaviruses that cause severe disease are frequent [16,17,18,19,20]. For these reasons, we are looking for affordable compounds to treat bunyavirus infections. Viral RNA polymerases (vRNApol) are adequate targets for DAAs, because these proteins are essential for viral genome replication and share sequence as well as structural similarities across different families of RNA viruses [20,21,22]. Structural studies of a known influenza endonuclease inhibitor (2,4-dioxo-4-phenylbutanoic acid or DPBA) bound to the endonuclease domain of La Crosse bunyavirus (LACV) vRNApol, suggested that structure-based studies can identify compounds with antiviral activity against different RNA viruses [23]. DPBA is a known inhibitor of the influenza virus vRNPpol endonuclease domain that performs the cap-snatching process [24]. Bunyaviruses also perform cap-snatching to prime viral transcription. Basically, the vRNApol synthesizes viral messenger RNAs using short capped primers derived from cellular transcripts after endonuclease-mediated cleavage [25]. Structural and functional similarities between the endonucleases of influenza virus and bunyaviruses suggest that endonuclease inhibitors designed against the influenza virus endonuclease could be repurposed to target bunyaviruses [23,25,26].

To identify potential inhibitors of BUNV vRNApol, we looked for molecules already used to treat other diseases and performed an in silico study to identify compounds similar to DPBA that could bind to the vRNApol endonuclease domain of Hantaan virus, a bunyavirus belonging to hantavirus genus [27]. From a list of fifteen top candidates, we selected five compounds for functional studies in cell culture. One of them, acetylsalicylic acid, efficiently inhibited BUNV infection by blocking the biogenesis of the viral replication organelle.

## 2. Materials and Methods

### 2.1. Computational Models

The discovery of novel antivirals has initially been guided by two computational strategies. A ligand-based virtual screening was first performed by using BRUSELAS code [28]. In that approach, the structure of DPBA, a well-known inhibitor of the virus endonuclease reaction, is used as a chemical template for the search of similar compounds in the DrugBank (DB) database [29]. For the records, that similarity scheme is based on a fingerprint derived from a wide panel of numeric descriptors including molecular weight, number of rotatable bonds, number of hydrogen acceptor/donor atoms, topological polar surface area and octanol-water partition coefficient [28]. In a second stage, docking simulations with LeadFinder [30] as implemented in Flare [31] are conducted to screen DB against the crystal structure of the cap-snatching endonuclease of Hantaan virus, which is deposited in the Protein Data Bank with code 5IZE [27]. Docking allows assessment of the protein-ligand interaction by generating a series of ‘poses’ of the drug in the target binding site [32,33,34]. DB was selected as the working library because it compiles both Food and Drug Administration (FDA) approved drugs and experimental molecules and has been successfully used in drug repurposing [35,36].

### 2.2. Cells, Virus and Drugs

Vero cells (CCL-81) were obtained from the American Type Culture Collection and grown in DMEM (Sigma-Aldrich, St. Louis, MO, USA; D6429) supplemented with 10% fetal bovine serum (FBS, Reactiva S.A., Barcelona, Spain), 1% MEM nonessential amino acids (Sigma-Aldrich; M7145), 1% L-glutamine (Merck, Rahway, NJ, USA; 59202C) and 1% penicillin/streptomycin (Sigma-Aldrich; P4333).

Bunyamwera virus (BUNV, VR-87) was obtained from the American Type Culture Collection (Virginia, WV, USA). Propagation and production of BUNV stocks was performed in Vero cells as previously described [9,37]. Virus titers measured as plaque forming units, or PFU per milliliter, were determined by standard plaque assay in Vero cells using an agar overlay [37].

Acetylsalicylic acid (A5376), myricetin (M6760), L-phenylalanine (5482), p-aminohippuric acid (A1422) and silibinin A (S0417) were purchased from Sigma-Aldrich, dissolved in ethanol (acetylsalicylic acid and myricetin), water (L-phenylalanine and p-aminohippuric acid) or dimethyl sulfoxide (DMSO) (silibinin A), as recommended by the manufacturer, to prepare stock solutions and stored at −20 °C.

### 2.3. Cytotoxicity Assays

Cytotoxicity of the drugs was analyzed with the MTT (3-[4,5-dimethylthiazol-2-yl]-2,5 diphenyl tetrazolium bromide) assay, which measures the mitochondrial dehydrogenase activity of living cells [38]. Vero cells were seeded in a 96-well plate at a density of 10,000 cells per well and cultured until an 80% confluency was reached. Cells were incubated with serial dilutions of the drugs in DMEM for 24 h and then treated with MTT reagent (Sigma-Aldrich; M5655) at 0.5 mg/mL for 4 h. The formazan crystals were dissolved by incubation with the MTT solvent (10% SDS, 0.01 M HCl in 85% isopropanol) for 30 min in an orbital shaker protected from light. The amount of MTT formazan formed was determined spectroscopically at 570 nm in an ELISA plate reader. The background was also measured and subtracted at 690 nm. The cytotoxic effect of the drugs was measured with the CC_50_ calculated from dose–response curves. Three independent replicas were analyzed.

### 2.4. Viral Infection and Antiviral Activity

Vero cells were absorbed with BUNV at a multiplicity of infection (MOI) of 1 PFU per cell (viral stock diluted in DMEM culture medium supplemented to contain 2% FBS) at 37 °C for 1 h, gently moving the plate side to side every 15 min. After the inoculum was removed, fresh DMEM supplemented with 2% FBS was added, and cells were incubated for 8 h at 37 °C.

The effect of the compounds on the viral infection was determined by immunofluorescence and fluorescence microscopy. Vero cells were seeded in 96-well flat-bottom plates and incubated at 37 °C to reach 80–90% confluency. Cell monolayers were infected with BUNV at an MOI of 1 PFU/cell for 1 h at 37 °C. The viral inoculum was then removed and a serial dilution of acetylsalicylic acid (ASA), myricetin, L-phenylalanine, p-aminohippuric acid and silibinin A diluted in DMEM supplemented with 2% FBS was added. Infected cells not treated with drugs were also analyzed as controls. After 8 h, cells were fixed with 4% paraformaldehyde (PFA, TAAB Laboratories, Berks, England) for 20 min and washed three times with PBS. The percentage of infected cells was determined by immunofluorescence with an antibody specific for the viral N protein and fluorescence microscopy (see below). To analyze the recovery of BUNV after removing ASA from cell cultures, Vero cells were absorbed with BUNV at an MOI of 1 PFU/cell for 1 h and treated with ASA 10 mM for 8 h. The medium was removed and replaced by new DMEM with 2% FBS. After incubation at different time points (2, 4, 6 and 8 h), cells were fixed for 20 min with 4% PFA, washed three times with PBS and processed for immunofluorescence with an antibody specific for the viral N protein and studied by fluorescence microscopy. Images were acquired using a Leica DMi8 S widefield epifluorescence microscope and analyzed with the Image J software. The IC_50_ was calculated from the percentage of infected cells. Inhibition data were plotted as dose–effect curves fitted to a nonlinear regression model using the GraphPad Prism 9.5.0 software. All experiments were replicated three times.

### 2.5. Western Blot

Vero cells were infected with BUNV at 1 PFU/cell and treated with ASA at different doses (5, 10 and 15 mM) in 6-well plates. Non-infected cells were also processed as controls. Eight hours after infection, cells were harvested with CellStripper™ reagent (Corning^®^, Somerville, MA, USA; 25-056-CI) and incubated with 50 μL of RIPA lysis buffer (Thermo Scientific™, Waltham, MA, USA; 89900) with protease inhibitors for 30 min on ice. The amount of protein in cell lysates was quantified to ensure equal loading (10 μg) in Western blot gels using Pierce™ BCA Protein Assay Kit (Thermo Scientific™; 23227) according to the manufacturer’s specifications. The samples were analyzed by SDS-PAGE and transferred to polyvinylidene fluoride (PVDF) membranes by standard blotting procedures. Membranes were blocked overnight at 4 °C with PBS containing non-fat dry milk and 0.05% Tween 20, incubated for 1 h at room temperature with a rabbit anti-BUNV polyclonal antiserum, kindly provided by Prof. Richard Elliott [37] diluted 1:2000 in blocking buffer and a mouse anti-α-tubulin monoclonal antibody (Sigma-Aldrich; T5168) diluted 1:1000 in blocking buffer. After washing with this buffer, membranes were incubated with horseradish peroxidase (HRP)-conjugated secondary antibodies for 1 h at room temperature (RT). The target proteins were detected by ECL solutions (SuperSignal™ West Dura Extended Duration Substrate, Thermo Scientific™) with a ChemiDoc Imager (BioRad). The band intensity of Gc and N viral proteins was calculated using Image Lab software (BioRad) and normalized to α-tubulin. Three independent replicas were analyzed.

### 2.6. Immunofluorescence and Confocal Microscopy

Cells were grown on sterile glass coverslips, fixed with 4% PFA, and washed three times with PBS. Fixed cells were permeabilized and blocked by incubation for 40 min with PBS containing 0.25% saponin and 2% FBS. Cells were incubated for 1 h with primary antibodies diluted 1/200 in PBS with saponin and FBS. The following antibodies were used: a rabbit polyclonal antibody specific for the BUNV nucleocapsid (N) protein [7], a mouse monoclonal antibody specific for BUNV Gc glycoprotein, kindly provided by Prof. Richard M. Elliott [39] and a rabbit anti-Giantin polyclonal antibody that is a marker of the Golgi complex (BioLegend, San Diego, CA, United States; 924302). Alexa Fluor–conjugated antibodies (Invitrogen) were used as secondary antibodies diluted 1/500 in PBS with 0.25% saponin and 2% FBS. Cell nuclei were stained with 4′,6-diamidino-2-phenylindole (DAPI) (Sigma-Aldrich) diluted 1/200 in PBS with 0.25% saponin and 2% FBS for 20 min. All incubations were conducted at RT. Coverslips were mounted using Prolong-Gold (Life Technologies, Carlsbad, CA, USA). Confocal microscopy images were acquired using a Leica STELLARIS 5 confocal multispectral microscope equipped with an HCX PL APO 63.0 X/1.4 NA oil objective and LAS X Life Science software (Leica Microsystems).

### 2.7. Transmission Electron Microscopy (TEM)

Cells were grown on sterile glass coverslips in 6-well plates at 37 °C to reach a 90% confluency, before infection and drug treatment. Vero cells were mock-infected or infected with BUNV at an MOI of 1 PFU/cell for 1 h at 37 °C, and incubated in the absence or presence of three different concentrations (5, 10 and 15 mM) of ASA and fixed at 8 hpi with 1% glutaraldehyde in 0.2 M Hepes buffer, pH 7.4, at RT for 2 h. Cells were washed twice with Hepes 0.4 M, pH 7.4, postfixed by incubation at 4 °C for 1 h with a mixture of 1% osmium tetroxide and 0.8% potassium ferricyanide in water and washed three times with Hepes 0.4 M. Cells were then dehydrated in 5-min steps with increasing concentrations of ethanol (25, 50, 70 and 95%) in 10-min steps, and twice in 100% ethanol at 4 °C. Samples were processed for flat embedding in the epoxy resin EML-812 (TAAB Laboratories) by incubating for 2 h with a 1:1 mixture of ethanol–resin at RT. Cells were infiltrated for 16 h in pure resin and polymerized at 60 °C for 48 h. Ultrathin (∼50–70 nm) oriented serial sections were obtained using an UC6 ultramicrotome (Leica Microsystems), collected on uncoated 300-mesh copper grids (TAAB Laboratories), stained with saturated uranyl acetate and lead citrate, and imaged by TEM. Images were acquired using a Jeol JEM 1400 Flash electron microscope operating at 120 kV and equipped with a Gatan CMOS Oneview camera. At least 100 cells per condition were studied by TEM.

### 2.8. Quantification and Statistical Analysis

Experiments were conducted with a minimum of three independent replicates. Data presented in the text are expressed as the mean ± SD. Statistical significance was determined using two-sample unequal variance student’s *t*-test with two-tailed distribution (α = 0.05). Graphs were prepared and statistical analyses were conducted using GraphPad Prism 9.5.0 software.

## 3. Results

### 3.1. Antiviral Selection by Molecular Modeling

As described above, our computational models combine both ligand-based and structure-based molecular models. The former is based on the DPBA inhibitor, while the latter uses the crystal structure of the endonuclease of Hantaan virus resolved by Reguera, Cusack and co-workers [27]. These authors demonstrated the essential role played not only by the catalytic His36, but also by other critical residues, e.g., Glu54, Asp97 and Lys124, as they are also involved in Mn^2+^ ion binding [27]. Virtual Screening with docking was consequently focused on that specific region. The performed simulations lead to a series of five hits: ASA, L-phenylalanine and p-aminohippuric acid were selected by similarity; myricetin and silibinin A are associated with the more negative (stable) binding poses in the catalytic pocket. Figure 1 summarizes predicted interactions energies and illustrates the binding mode at the endonuclease catalytic site.

### 3.2. Antiviral Activity in Cells Infected with BUNV

To evaluate the antiviral activity of the drugs, Vero cells were infected with BUNV for 1 h and treated in parallel with increasing concentrations of the selected compounds. Drug cytotoxicity was first measured using the MTT test, and safe drug doses were used to determine the percentage of infected cells by immunofluorescence, using an antibody specific for the BUNV N protein. Myricetin, L-phenylalanine, p-aminohippuric acid and silibinin A showed no antiviral activity. However, ASA presented antiviral activity against BUNV (Figure 2). The percentage of cells with an N viral protein fluorescence signal decreased in infected Vero cells treated with increasing concentrations of ASA (Figure 2A–E). By regression analysis, we calculated the concentration of compound required to inhibit 50% of virus infection (IC_50_), and the concentration for 50% cytotoxic effect (CC_50_). The graph in Figure 2F shows that ASA efficiently inhibited BUNV infection at non-toxic concentrations, with an IC_50_ of 2.02 mM. To study ASA antiviral capacity in detail, further assays were performed with this compound at 5, 10 and 15 mM.

### 3.3. Effect of ASA on Viral Titer and Viral Protein Expression

To study the action of ASA in the production of infectious viral particles, Vero cells were infected at an MOI of 1 PFU/cell for 1 h at 37 °C, and incubated in the absence or presence of 5, 10 or 15 mM of ASA. At 8 hpi, supernatants were collected and titrated by standard plaque assay [37]. ASA treatment dramatically reduced the number of lysis plaques (Figure 3A). The amount of infectious viral particles in cell culture supernatants was reduced by 10 to 1000-fold (Figure 3B).

The effect of ASA in the expression of BUNV proteins was measured by Western blot (WB) using a rabbit anti-BUNV polyclonal antiserum that mainly recognizes the nucleocapsid N protein and the large Gc glycoprotein of BUNV (Figure 4). The WB image on the left shows that expression of N and Gc viral proteins in Vero cells was significantly reduced with the ASA treatment. The table of relative frequencies of Figure 4 shows a dose-dependent reduction of N and Gc expression levels in cells treated with ASA compared to untreated cells. These results suggest that the decrease in the amount of infectious viral particles in cell supernatants from ASA-treated cells shown in Figure 3, is due to the blockade of an early step of BUNV infection.

### 3.4. Distribution of Viral Gc and N Proteins in Cells Treated with ASA

BUNV Gc glycoprotein is a marker of viral factories that assemble in the Golgi complex [9]. BUNV N is a multifunctional protein that is synthesized in large amounts early in infection and accumulates in the cytosol and in association with cell membranes [7,40]. To study the effects of ASA in Gc and N localization in cells, we processed infected cells, treated or untreated with ASA by immunofluorescence using antibodies specific for Gc, N and the Golgi marker Giantin. As mentioned before, the number of infected cells decreased with ASA treatment. The percentage of cells with N positive signal was 48%, 22% and 8% of cells treated with ASA 5, 10 and 15 mM, respectively. A Gc positive signal was presented in 22%, 14% and 1% of cells treated with ASA 5, 10 and 15 mM, respectively. Confocal microscopy showed that the fluorescence signal of both viral proteins decreased in a dose-dependent manner (Figure 2A–D and Figure 5A–D). Moreover, ASA inhibited the characteristic BUNV-induced Golgi fragmentation and dispersion. For comparison, the normal morphology and intracellular localization of the Golgi complex in uninfected Vero cells non-treated with ASA are shown in Appendix A. With ASA, Gc was retained in perinuclear areas where it colocalized with Golgi (Figure 5A–D). The number of cells with N fluorescent signal as well as the intensity of the signal decreased with increasing concentrations of ASA, but the cytosolic distribution of N protein did not change in the presence of ASA (Figure 2A–D). Three-dimensional reconstructions (Figure 5E,F) showed that in non-treated cells, Gc is dispersed in the cytosol where it associates to fragments of Golgi (Figure 5E) and to virions inside secretory vesicles [9]. With ASA, both Gc and Giantin signals grouped in the perinuclear region (arrows in Figure 5F). With 5 mM ASA, no significant differences were observed in the number of cells with N and Gc protein signals; however, there was a significant reduction with the 10 and 15 mM treatments (Figure 5G). Quantification of Gc distribution in infected cells not treated and treated with 5, 10 or 15 mM of ASA, is shown in Figure 5H that shows two phenotypes corresponding to scattered or perinuclear distribution of Gc in the cytosol (Figure 5H). This analysis revealed significant differences in the localization of Gc signal in infected cells without ASA or in cells treated with 5 mM ASA, compared to cells treated with higher ASA doses.

### 3.5. Effect of ASA on the Morphology of Viral Structures and Cellular Organelles

To further investigate the mechanism of action of ASA, we studied the ultrastructure of infected Vero cells not treated or treated with ASA (5, 10 and 15 mM) by transmission electron microscopy (Figure 6). Infected cells not treated with the drug showed a recruitment of cellular organelles to particular areas of the cytoplasm forming the viral factory (VF), structure that contains elements of the Golgi complex, endoplasmic reticulum (ER) and mitochondria (Figure 6A,B). Extracellular virions were also seen close to the plasma membrane of infected cells (inset in Figure 6A). High magnification images of VFs show characteristic spherules, tubes and viral particles (Figure 6C).

In infected cells treated with 5 mM ASA, cellular organelles were not apparently recruited to a specific area into the cytoplasm forming a factory (Figure 6D). However, some infected and ASA-treated cells showed the BUNV factory (Figure 6E) with spherules in Golgi membranes (Figure 6F). In infected cells treated with 10 mM ASA, the cellular organelles were randomly distributed through the cytoplasm and no characteristic VFs were seen (Figure 6G). Higher magnification images showed altered Golgi complexes with swollen sacculi, and neither spherules nor viral particles were seen in the majority of cells (Figure 6H,I). Similar results were obtained with infected cells treated with 15 mM ASA (Figure 6J–L). Quantification of viral spherules, intracellular and extracellular viral particles is shown in Figure 7. Tables of frequencies and graphs show that ASA treatment remarkably reduced the number of spherules and viral particles, being practically undetectable with the treatment of 15 mM ASA. We also analyzed the effect of the ASA treatment in the ultrastructure of uninfected Vero cells (Appendix A). The distribution of cellular organelles was not modified in cells treated with lower or higher concentrations of ASA in comparison to untreated cells (Appendix A). Closer views of the Golgi complex, ER and mitochondria did not show visible alterations of those cellular organelles with any of the ASA treatments (Appendix A). High magnification views of the Golgi complex showed that Golgi sacculi were not swollen (Appendix A), a feature detected only in BUNV-infected and ASA-treated cells. These results suggest that ASA treatment interferes with BUNV infection by blocking the formation of viral spherules.

### 3.6. Recovery of Cell Morphology and BUNV Infection after Elimination of ASA

To evaluate the capacity of BUNV to recover infection after removing ASA from cell cultures, Vero cells were infected and treated with 10 mM ASA, and 8 h later the medium was replaced by fresh DMEM with 2% FBS. Cells were then incubated for 0, 2, 4, 6 or 8 h without ASA and processed for immunofluorescence and fluorescence microscopy, titration by plaque assay and electron microscopy. The percentage of infected cells increased after removing the drug and reached values around 90–100% of infected cells after 8 h of incubation without ASA (Figure 8A). In addition, infectious viral particles were detected by plaque assay after incubation without drug (Figure 8B). Electron microscopy of infected cells treated with ASA showed Golgi complexes with swollen sacculi and no viral structures (Figure 8C). After incubation without drug for 2, 4, 6 or 8 h, the Golgi complex had already recovered its normal morphology and showed viral spherules and viral particles (Figure 8D–H). Therefore, after the elimination of ASA, the Golgi complex recovered and BUNV was able to induce a productive infection.

## 4. Discussion

In our search for antivirals to combat bunyavirus infections, we have discovered that ASA inhibits the infection of a bunyavirus by interfering with the biogenesis of its replication organelle. ASA is on the WHO model list of essential medicines [41] that includes the most efficacious, safe and cost–effective medicines for priority conditions. In addition, ASA is one of the most widely used medications in the world to treat pain, rheumatic fever, arthritis and inflammation [42]. A potential use of ASA to reduce the long-term risk of several cancers has been proposed [42,43,44,45] and ASA’s antiviral properties have also been reported. ASA inhibits the replication of flaviviruses, such as Japanese encephalitis virus (JEV), dengue virus (DENV) and hepatitis C virus (HCV) [46,47]. ASA also blocks the propagation of respiratory viruses such as influenza A virus and the human rhinovirus (HRV) [48].

Structural and functional similarities between the endonucleases of the influenza virus and bunyaviruses suggested that endonuclease inhibitors designed against influenza virus endonuclease could be repurposed to target bunyaviruses [23,25,26]. That earlier evidence motivated the present screening of the DrugBank database, a library that contains ca. 10,000 compounds previously used to treat humans, including both approved and experimental drugs [29]. Our hypothesis was that a similar drug to DPBA, a known inhibitor of influenza virus endonuclease [27], might also exhort a biological activity against bunyaviruses. That search was not manually conducted but guided by computational methods that used fingerprints to systematically assess similarities. Virtual screening also confirms the compatibility with the catalytic site, a prerequisite to halt the endonuclease machinery.

The antiviral properties of diketoacid inhibitors such as DPBA have been tested in cells infected with different viruses such as the influenza virus [49], the human immunodeficiency virus 1 (HIV-1) [50,51] and bunyaviruses [52,53]. To our knowledge, no clinical trials for diketoacid inhibitors to treat virus disease have been initiated.

Here we show that ASA induced a decrease of up to three logarithmic units in viral titers of culture supernatants from BUNV-infected Vero cells. It also produced a significant dose-dependent reduction of the expression levels of Gc and N viral proteins and inhibited the assembly of Golgi-associated BUNV spherules and new viral particles. Light and electron microscopy showed that ASA protected the Golgi complex from the characteristic BUNV-induced fragmentation and caused swelling of Golgi sacculi in BUNV-infected cells only, while in mock-infected cells treated with ASA the Golgi complex looked normal. These results suggest that the way vRNApol molecules interact with Golgi membranes changes when bound to ASA, something that probably impairs the structural role of vRNApol in the biogenesis of viral replication organelles and VRCs. Similar results were obtained in Vero cells infected with BUNV and treated with the typical vRNApol inhibitor Ribavirin where viral spherules were absent or disrupted [54].

The elimination of ASA from the cell culture medium led to the recovery of the normal morphology of Golgi and of all viral structures, such as spherules (ROs), intracellular and extracellular viral particles. A positive aspect of these results is that the secondary effects in cell morphology (swollen Golgi) did not affect cell viability (Figure 2) and were reversible (Figure 8). However, the reversibility of the antiviral effect has to be also taken into consideration for stablishing the most adequate treatment conditions in experiments with animals, such as number of doses over time. When using therapeutic doses, plasma levels of ASA can range from 0.2 to 0.5 mM and the toxic level is stablished above 8 mM [55], which is clearly higher than the concentration of 2 mM (IC_50_) that efficiently inhibited BUNV replication in cell culture. However, due to its anti-coagulant properties, ASA could not be used to treat patients with hemorrhagic fever. Examples of bunyavirus infections that could be treated with ASA and others that should not be treated with ASA are shown in Table 1. ASA is a cheap and safe drug with the potential to become a broad-spectrum antiviral.

## Figures and Tables

**Figure 1 viruses-15-00948-f001:**
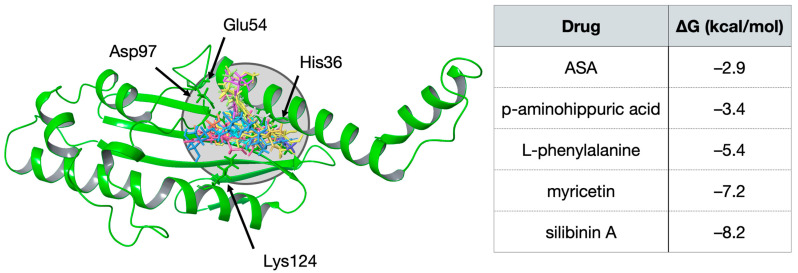
**Molecular modeling and summary of the numeric outcomes**. (**Left** panel): poses adopted by the selected compounds at the endonuclease catalytic site (drugs are represented as colored wireframes and target is sketched in green cartoons). Main residues are indicated by arrows. (**Right** panel): Computed free energies for the drug–endonuclease complexes (in kcal/mol).

**Figure 2 viruses-15-00948-f002:**
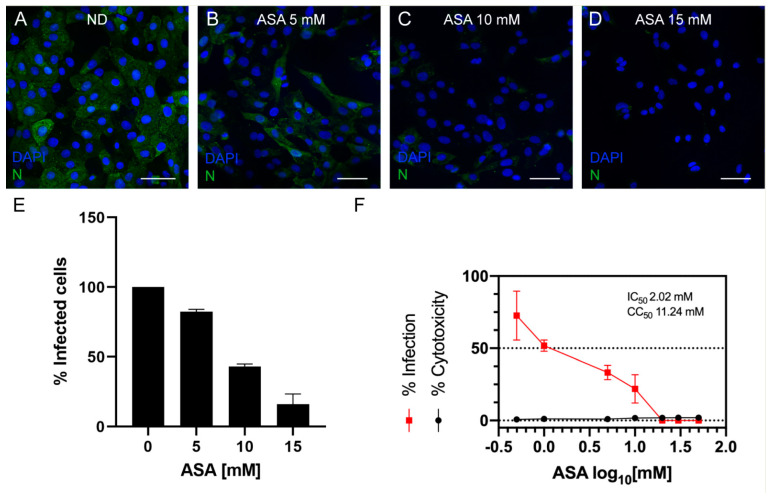
**Antiviral activity of ASA against BUNV**. Vero cells were absorbed 1 h with BUNV at an MOI of 1 PFU/cell, exposed to increasing concentrations of ASA for 8 h and processed by immunofluorescence. (**A**–**D**) Fluorescence microscopy of infected cells untreated (**A**) and treated with 5, 10 and 15 mM ASA (**B**–**D**). Nuclei are labeled with DAPI (blue) and viral infection with an antibody specific for BUNV nucleoprotein (N) (green). (**E**) Percentage of infected cells measured by immunofluorescence in the absence of ASA treatment and with three different concentrations of ASA. Data are shown as mean ± s.e.m. of three independent experiments. (**F**) Dose–response curve (red line) of ASA was determined by nonlinear regression. Cytotoxic effect on Vero cells exposed to increasing concentrations of the drug in the absence of virus is also shown (black line). IC_50_ and CC_50_ were determined from dose–response curves based on treatment with seven different concentrations. IC_50_ was calculated from the percentage of infected cells, and CC_50_ was determined by the MTT (3-(4,5-dimethylthiazol-2-yl)-2,5-diphenyltetrazolium bromide) cell viability assay. ND, no drug. Scale bars: 50 μm.

**Figure 3 viruses-15-00948-f003:**
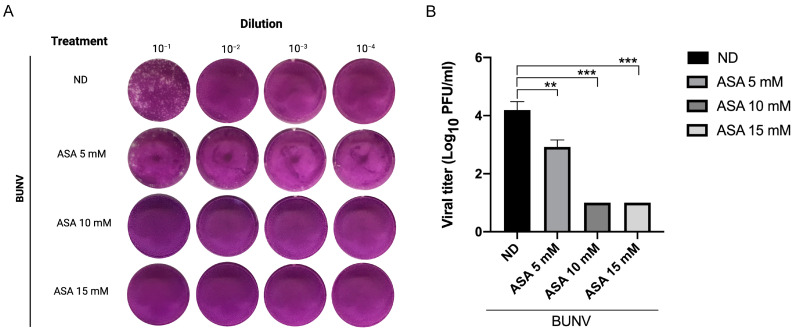
**Study of ASA antiviral effect by plaque reduction assay**. BUNV was propagated in Vero cells in the absence or presence of different concentrations of ASA (5, 10 and 15 mM) for 8 h, the supernatants were collected and the virus titer was tested by plaque assay. (**A**) A representative picture showing the plaque reduction assay with the drug. (**B**) BUNV titers from cell supernatants without ASA and with three different concentrations of ASA. Each bar shows the mean value ± s.e.m. Significance was determined by unpaired two-tailed student’s *t*-test; ns indicates “not significant”, ** *p* < 0.01 and *** *p* < 0.001.

**Figure 4 viruses-15-00948-f004:**
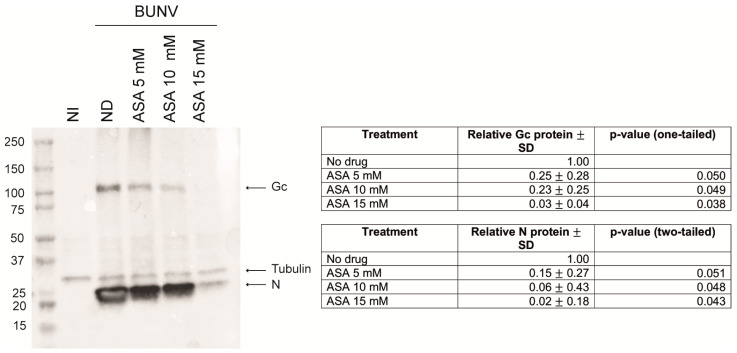
**Effect of ASA on viral protein expression**. Non-infected and BUNV-infected Vero cells at an MOI of 1 PFU/cell for 1 h and incubated in the presence or absence of ASA (5, 10 and 15 mM) for 8 h, were processed for Western blot analysis. An amount of 10 micrograms of the cell lysates were loaded and samples were probed with an antiserum against BUNV. The tubulin signal was used as loading control. NI, non-infected; ND, no drug. One experiment representative of 3 is shown. The tables show the average relative quantity of the Gc protein and N protein ± standard deviation (SD) for all the experiments. The *p*-values (one- and two-tailed, respectively) were calculated with the student’s *t*-test.

**Figure 5 viruses-15-00948-f005:**
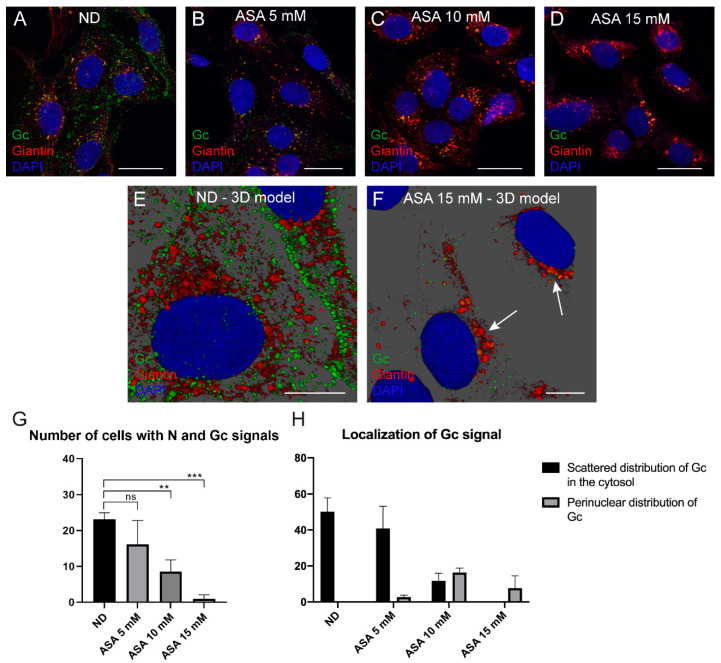
**ASA treatment induces the accumulation of BUNV Gc glycoprotein in the cell perinuclear region**. Vero cells were absorbed 1 h with BUNV at an MOI of 1 PFU/cell, incubated in the presence or absence of ASA (5, 10 and 15 mM) for 8 h and processed by immunofluorescence. (**A**–**F**) Confocal fluorescence microscopy of infected cells untreated and treated with ASA. The nucleus is labeled with DAPI (blue), the BUNV Gc glycoprotein (green) with a specific antibody, and the Golgi complex with the anti-Giantin antibody (red). In untreated cells (**A**), Gc signal is distributed throughout the cytoplasm, while Gc signal in ASA-treated cells (**B**–**D**) is retained in a perinuclear region associated with the Golgi complex. (**E**,**F**) Three-dimensional reconstructions from a series of confocal images. In the absence of the drug (**E**), Gc signal is dispersed, while in cells treated with 15 mM ASA (**F**), Gc signal accumulates near the nucleus in the Golgi complex (white arrows). (**G**) Average of the number of cells with both N and Gc viral proteins-associated fluorescent signals in the absence and presence of ASA measured by immunofluorescence. This average was determined by counting the number of cells with both N and Gc protein signals. (**H**) Quantification of the distribution of Gc IF signal shows that ASA induces a retention of Gc signal in the perinuclear region. Quantification was performed by counting the number of cells with Gc protein signal in the cytosol or in the perinuclear region. A total of 50 cells per condition were studied. The results in panels G and H are presented as the mean of the number of cells with viral proteins-associated fluorescence signals ± s.e.m. of three independent experiments. Significance was determined by unpaired two-tailed student’s *t*-test; ns indicates “not significant”, ** *p* < 0.01 and *** *p* < 0.001. ND, no drug. Scale bars: 25 μm in (**A**–**D**); 10 μm in (**E**,**F**).

**Figure 6 viruses-15-00948-f006:**
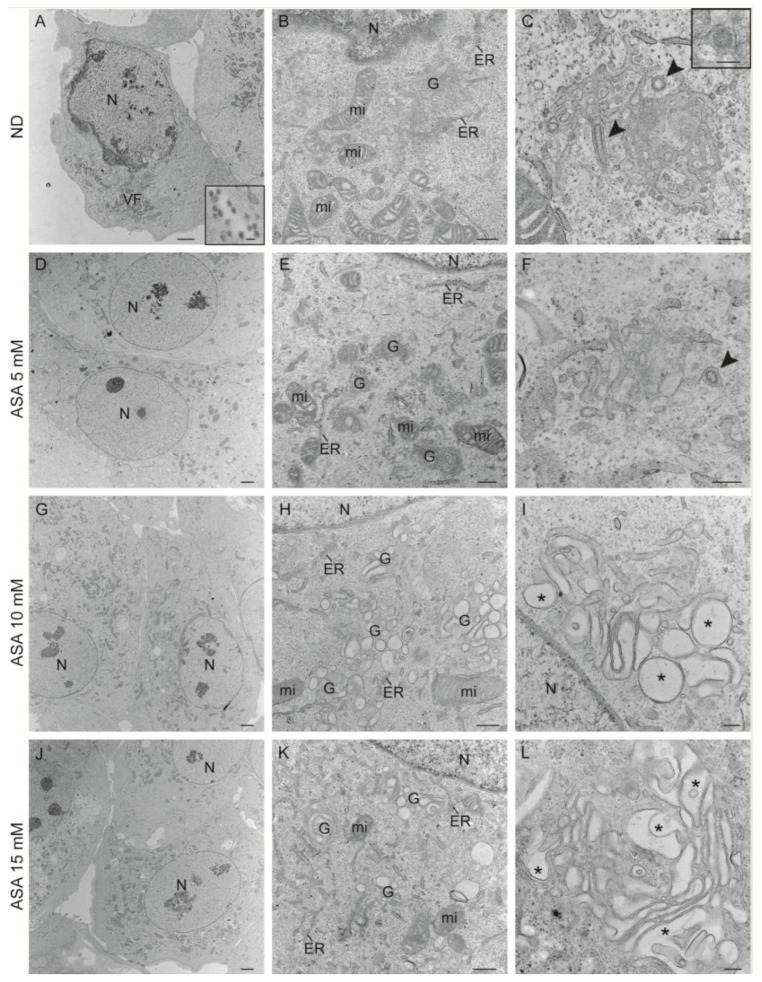
**Transmission electron microscopy of BUNV-infected cells untreated or treated with ASA**. Vero cells were absorbed 1 h with BUNV at an MOI of 1 PFU/cell, incubated in the presence or absence of ASA (5, 10 and 15 mM) for 8 h and processed by transmission electron microscopy. (**A**–**C**) BUNV-infected Vero cells not treated with ASA (no drug or ND). (**A**) Recruitment of cellular organelles to a discrete area of the cytoplasm forming the viral factory (VF). The inset shows extracellular virions. (**B**) Higher magnification view of the VF shows the Golgi complex (G) surrounded by mitochondria (mi) and endoplasmic reticulum (ER) membranes. Mitochondria have swollen cristae. (**C**) Bunyavirus replication organelles (spherules and tubules, arrowheads) in Golgi membranes. The inset shows an intracellular virus in the Golgi complex. (**D**–**F**) BUNV-infected Vero cells treated with ASA 5 mM. The cellular organelles are not recruited to a specific area into the cytoplasm (**D**). Golgi complex stacks surrounded by endoplasmic reticulum membranes and modified mitochondria (**E**). The arrowhead in (**F**) points to a viral spherule in Golgi membranes. (**G**–**L**) BUNV-infected Vero cells treated with ASA 10 or 15 mM. Low magnification images (**G**,**J**) show cellular organelles distributed through the cytoplasm not forming a characteristic VF. Golgi membranes are swollen without bunyavirus spherules and mitochondria have normal morphology and shape (**H**,**K**). Details of modified Golgi complex (asterisks) are shown in (**I**,**L**). Neither BUNV particles nor spherules were detected. N, nucleus. Scale bars: 2 μm in (**A**,**D**,**G**,**J**); 500 nm in (**B**,**E**,**H**,**K**); 200 nm in the inset of (**A**,**C**,**F**,**I**,**L**); 100 nm in the inset of (**C**).

**Figure 7 viruses-15-00948-f007:**
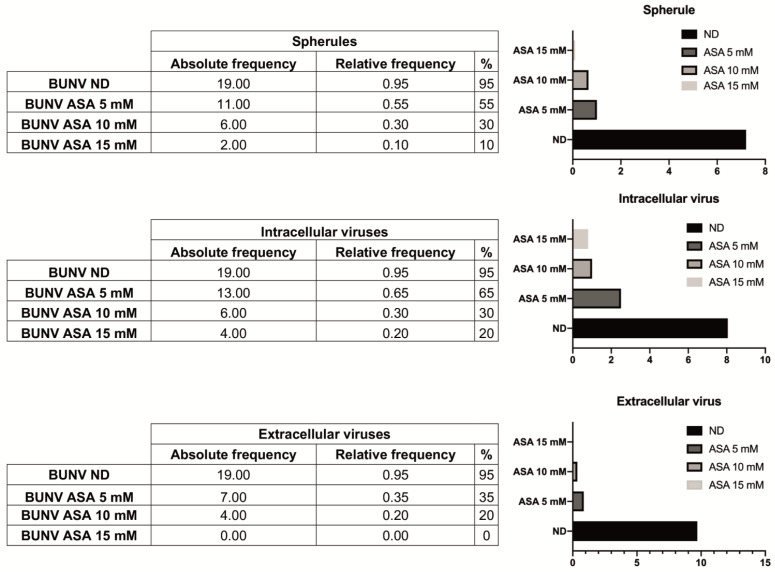
Quantification of spherules, intracellular and extracellular viruses in BUNV-infected cells, with or without ASA using transmission electron microscopy. Vero cells were absorbed 1 h with BUNV at an MOI of 1 PFU/cell, incubated in the presence or absence of ASA (5, 10 and 15 mM) for 8 h and processed by embedding in epoxy resin, ultramicrotomy and electron microscopy. The table shows the absolute and relative frequencies of spherules, intracellular and extracellular viruses counted in BUNV-infected cells in each condition: BUNV ND (infected with no ASA), BUNV + ASA 5 mM, BUNV + ASA 10 mM and BUNV + ASA 15 mM. ASA treatment produces a significant reduction of the number of viral structures. The graphs showing the means of spherules, intracellular and extracellular viruses, counted in a total of 20 cells per condition, are shown near each table, on the right.

**Figure 8 viruses-15-00948-f008:**
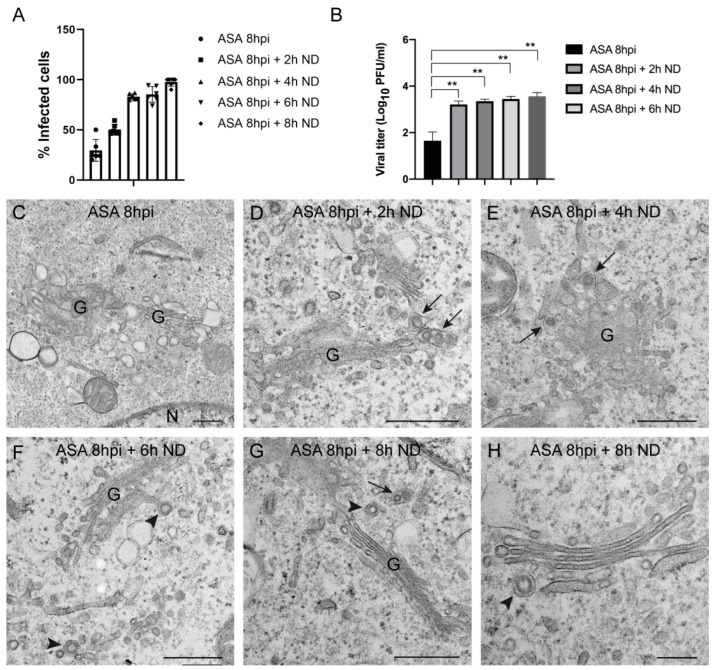
**Recovery of a productive BUNV infection after elimination of ASA from cell cultures.** Vero cells were absorbed 1 h with BUNV at an MOI of 1 PFU/cell and treated with ASA 10 mM for 8 h. The medium was removed and replaced by new DMEM with 2% FBS. After incubation at different time points (2, 4, 6 and 8 h), cells were processed for immunofluorescence, titration by plaque assay and electron microscopy. (**A**) The percentage of infected cells per condition was determined by fluorescence microscopy using an antibody specific for BUNV N protein. After removing the drug from the cell culture medium, the number of infected cells increased over time. (**B**) Graph showing the viral titer at different time points. Each bar shows the mean value ± s.e.m. Significance was determined by unpaired two-tailed student’s *t*-test, where ** indicates *p* < 0.01. (**C**–**H**) Electron microscopy study. After 8 h of ASA treatment (**C**), the Golgi membranes were swollen. Neither BUNV spherules nor viral particles were detected in these infected and ASA-treated cells. (**D**–**G**) After ASA elimination, BUNV particles (arrows) and spherules (arrowheads) are seen associated with Golgi membranes. (**H**) High magnification image of a Golgi complex with a viral spherule (arrowhead). N, nucleus; G, Golgi complex. Scale bars: 500 nm in (**C**–**G**); 200 nm in (**H**).

**Table 1 viruses-15-00948-t001:** Examples of members of the *Bunyavirales* order and associated diseases that could potentially be treated with ASA, assuming that studies with animal models and clinical trials will give good results, and those that could not be treated with ASA because they produce hemorrhagic fever. Distribution, reservoirs and vectors are included. N.I., not identified. Sources: CDC, NIH and ECDC.

Virus	Disease	Distribution	Reservoir	Vector	ASA
**La Crosse**	Childhood encephalitis	North, Central and South America	Small mammals (chipmunks and squirrels)	Mosquitoes	Yes
**Cache Valley** **virus**	Fever, encephalitis, meningitis	North and Central America	Deer, cattle, horses and sheep	N.I.	Yes
**Toscana virus**	Meningitis, encephalitis	Mediterranean region of Western Europe (Portugal, Spain, France, Greece and Croatia), Cyprus and Turkey	Possibly migratory birds and domestic animals	Sandflies	Yes
**Uukuniemi virus**	Fever, headache, muscle and joint pain	Scandinavia	N.I.	Ticks	Yes
**Schmallenberg**	Congenital malformations and abortion in livestock	Belgium, Germany, France, Italy, Luxemburg, the Netherlands, Spain and United Kingdom	N.I.	Possibly mosquitoes and midges	Yes
**RVFV**	Acute and febrile illness with few severe cases (neurological disorders, partial or complete blindness, hemorrhagic fever or thrombosis) High mortality and abortion in domestic ruminants	Africa, Saudi Arabia and Yemen	Livestock (cattle, sheep, goats, buffalos and camels)	Mosquitoes	No
**CCHFV**	Hemorrhagic fever	Eastern and Southern Europe, Northwestern China, Africa, the Middle East and the Indian subcontinent	Wild and domestic animals	Ticks	No
**Lassa Fever virus**	Hemorrhagic fever	Endemic in parts of West Africa including Sierra Leone, Liberia, Guinea and Nigeria	Multimammate rat	Multimammate rat	No
**SFTSV** **(Dabie bandavirus)**	Hemorrhagic fever Severe fever with thrombocytopenia	Northwest and central China, Japan, South Korea, Vietnam and Taiwan	N.I.	Ticks	No
**Hantaan**	Hemorrhagic fever with renal syndrome (Old World) Hantavirus pulmonary syndrome (New World)	Europe and Asia North and South America	Rodents	Rodents	No

## Data Availability

Data will be available from the corresponding author at reasonable request.

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
