# Peer review of "Antiviral Activity of Acetylsalicylic Acid against Bunyamwera Virus in Cell Culture"

_viruses, 2023, doi:10.3390/v15040948_

Round 1

Reviewer 1 Report

In the present research article, Fernández-Sánchez et al use an in silico system to find potential inhibitors for Bunyamwera virus (BUNV) infection. Although, is not explained the actual state of art, of inhibitors and vaccines for this family of viruses. Why is important to find new antiviral compounds. For this study, they used in silico approach to find compounds that would bind to Hantaan virus RNA pol endonuclease domain. They selected 5 compounds from which only one was reducing viral particle production, Acetylsalicylic acid (ASA). Through transmission electronic microscopy (TEM) analysis, the authors concluded that ASA inhibits virus assembly by “affecting the biogenesis of the viral replication organelle” which is the Golgi apparatus. This is also depicted in the tittle.

The manuscript presents basic concepts problems in virology and lack of controls leading to misinterpretations that are indicated as follows.

The following revisions are needed

The major problem of this manuscript is to assume that ASA is affecting the biogenesis of the Golgi. In this context, there are only descriptive results that show altered Golgi morphology without a proper control that shows a normal Golgi morphology that is without infection. The Golgi of an infected cell, is not the correct control. Also, a mock control infected with ASA treatment is needed in order to observe whether the effect of ASA in the Golgi is related to infection or just a general effect on the organelle. There are no functional assays to address whether this altered morphology is affecting only viral proteins or an indirect effect. There are no experiments to show that the Golgi altered morphology is functionally affecting viral particle production. Therefore, the tittle should be edited.

Introduction

Line 46, not all the Bunyavirales order are transmitted through arthropods, also Hantaviruses belong to this order and are transmitted through rodents. Also, in this sentence and 47, the words are linked to Wikipedia which shows copy and paste. I find it a lack of respect.

Paragraph from line 45 to 59 have different format and letter size. Also from line 73 to 76

Line 79-80, Hantaan virus is not a bunyavirus, is a hantavirus, genus Orthohantavirus and Bunyavirales order.

Results

Figure 3 A, is very difficult to see the plaques in those images.

Line 307-308, the authors make analysis of N distribution from Figure 2 A-D but is it not shown what they are stating.

Line 310, the authors state assure about observing virions inside vesicles, how do they show this?

Figure 5, A ND image, the distribution of the Golgi is almost absent therefore the conclusions about the Golgi distribution are not credible. Also, from A to D it is clear that Gc is always localized at the Golgi, what it changes is the distribution outside de Golgi. From the pictures, is not clear that Gc is less in the Golgi with ASA treatment. A quantification of cells should be made to make that conclusion.

Figure G and H, what is the Y axis?

Figure 6, missing controls, mock infected and mock infected with ASA at least in supplementary. The zooms of the images in this figure are different, it is difficult to compare them.

An experiment to address whether there is an effect on the viral polymerase would be to check viral RNA with qPCR, because the authors only study viral proteins. ASA could be acting on translation but it is not known there is a direct effect on the polymerase.

Table 1 is not contributing to the paper. The authors cannot assume the effect of a drug according to the disease they produce, or reservoir. That is not correct, it only can be tested for the virus itself. Again, Hantaan virus is on the table and is not a bunyavirus, is a hantavirus.

Author Response

Reviewer 1

In the present research article, Fernández-Sánchez et al use an in silico system to find potential inhibitors for Bunyamwera virus (BUNV) infection. Although, is not explained the actual state of art, of inhibitors and vaccines for this family of viruses. Why is important to find new antiviral compounds.

Response. The first paragraph of Introduction describes: “However, no specific antivirals have been approved so far to treat bunyavirus infections. Ribavirin and Favipiravir, two inhibitors of viral RNA polymerases are currently being studied in ongoing clinical trials for Lassa fever and Crimean-Congo hemorrhagic fever (https://clinicaltrials.gov/)”.

We have modified the phrase: “However, there are no vaccines used worldwide and no specific antivirals have been approved so far to treat bunyavirus infections” (lines 41-42).

For this study, they used in silico approach to find compounds that would bind to Hantaan virus RNA pol endonuclease domain. They selected 5 compounds from which only one was reducing viral particle production, Acetylsalicylic acid (ASA). Through transmission electronic microscopy (TEM) analysis, the authors concluded that ASA inhibits virus assembly by “affecting the biogenesis of the viral replication organelle” which is the Golgi apparatus. This is also depicted in the tittle.

Response. We believe the reviewer is confused with the terms used in the manuscript. As described in the second paragraph of Introduction, the viral replication organelle of BUNV is not the Golgi apparatus but single-membrane vesicles or spherules assembled in Golgi membranes.

The manuscript presents basic concepts problems in virology and lack of controls leading to misinterpretations that are indicated as follows.

The following revisions are needed

The major problem of this manuscript is to assume that ASA is affecting the biogenesis of the Golgi. In this context, there are only descriptive results that show altered Golgi morphology without a proper control that shows a normal Golgi morphology that is without infection. The Golgi of an infected cell, is not the correct control. Also, a mock control infected with ASA treatment is needed in order to observe whether the effect of ASA in the Golgi is related to infection or just a general effect on the organelle.

Response. The requested controls were already included in the first version of the manuscript. Perhaps the reviewer did not see the supplementary Figure S1 (now Figure S2). Also, we do not say that ASA affects the biogenesis of Golgi but that of spherules (replication organelles of BUNV) that are detected in low amounts in BUNV-infected cells treated with ASA (Figures 6 and 7).

There are no functional assays to address whether this altered morphology is affecting only viral proteins or an indirect effect. There are no experiments to show that the Golgi altered morphology is functionally affecting viral particle production. Therefore, the tittle should be edited.

Response. The fact that the Golgi complex is swollen in BUNV-infected cells treated with ASA and not in mock-infected cells treated with ASA suggests that the interaction of ASA with viral proteins, presumably the vRNApol mediates this effect (lines 463-471).

Introduction

Line 46, not all the Bunyavirales order are transmitted through arthropods, also Hantaviruses belong to this order and are transmitted through rodents. Also, in this sentence and 47, the words are linked to Wikipedia which shows copy and paste. I find it a lack of respect.

Response. In the second paragraph of Introduction we stated: “The Bunyavirales order comprises more than 500 viruses that are generally defined as bunyaviruses. With the exception of Hantaviruses and Arenaviruses, all viruses in the Bunyavirales order are transmitted by arthropods [6].” Thus, we already mentioned that not all members of the Bunyavirales order are transmitted by arthropods. We do not know why the reviewer says that this statement comes from Wikipedia because it does not. The general information comes from many review such as the one we cite (Horne and Vanlandingham, Viruses 2014, reference number 6).

Paragraph from line 45 to 59 have different format and letter size. Also from line 73 to 76

Line 79-80, Hantaan virus is not a bunyavirus, is a hantavirus, genus Orthohantavirus and Bunyavirales order.

Response. In the second paragraph of Introduction we stated: “The Bunyavirales order comprises more than 500 viruses that are generally defined as bunyaviruses”. For simplification we use the term “bunyaviruses” that is generally accepted in the field.

Results

Figure 3 A, is very difficult to see the plaques in those images.

Response. In our files (both pdf and tiff) we clearly see the plaques in infected cells without ASA and in infected cells with ASA 5 mM.

Line 307-308, the authors make analysis of N distribution from Figure 2 A-D but is it not shown what they are stating.

Response. We have modified the phrase: “The intracellular distribution pattern of N did not change with ASA (Figure 2A-D)” and now it states: “The number of cells with N fluorescent signal as well as the intensity of the signal decreased with increasing concentrations of ASA, but the intracellular, mainly cytosolic distribution of N protein did not change in the presence of ASA (Figure 2A-D)”. Please, zoom the images in Figure 2 to appreciate the result.

Line 310, the authors state assure about observing virions inside vesicles, how do they show this?

Response. From previous studies, it is known that at 8 hpi, the cytosolic Gc-associated fluorescent puncta correspond to virions inside secretory vesicles on the way to the plasma membrane. We provide a reference (9. Salanueva et al., J. Virol. 2003) to document the observation.

Figure 5, A ND image, the distribution of the Golgi is almost absent therefore the conclusions about the Golgi distribution are not credible. Also, from A to D it is clear that Gc is always localized at the Golgi, what it changes is the distribution outside de Golgi. From the pictures, is not clear that Gc is less in the Golgi with ASA treatment. A quantification of cells should be made to make that conclusion.

Response. Our results show that ASA protects the Golgi complex from BUNV-induced fragmentation and dispersion (see giantin signals without and with ASA). To help with interpretation, we have added a new Figure S1 showing how the Golgi complex looks like in mock-infected, ND cells.

Figure G and H, what is the Y axis?

Response. The Y axis show the average of the number of cells with N and Gc fluorescent signals (from three independent experiments). This information is now included in figure and legend.

Figure 6, missing controls, mock infected and mock infected with ASA at least in supplementary.

Response. Figure 6 shows TEM images of BUNV-infected cells in the absence and presence of ASA. Figure S2 (Figure S1 in the previous version of the manuscript) shows TEM images of mock-infected cells in the absence and presence of ASA. We believe that the reviewer did not see this figure.

The zooms of the images in this figure are different, it is difficult to compare them.

Response. We are not sure about what the reviewer means. In Figures 6 and S1, we show images with three different magnifications per condition. To facilitate comparison, in all conditions, these three different magnifications are the same.

An experiment to address whether there is an effect on the viral polymerase would be to check viral RNA with qPCR, because the authors only study viral proteins. ASA could be acting on translation but it is not known there is a direct effect on the polymerase.

Response. Thank you for the suggestion. This is indeed a good experiment to study the mechanism of action of ASA in BUNV-infected cells. Considering that ASA was selected in silico as a potential inhibitor of the vRNApol, we believe that an action on this protein is possible, but of course we cannot exclude additional targets. This will be part of our future studies that will also investigate why the RO of the virus is not assembled in the presence of ASA.

Table 1 is not contributing to the paper. The authors cannot assume the effect of a drug according to the disease they produce, or reservoir. That is not correct, it only can be tested for the virus itself. Again, Hantaan virus is on the table and is not a bunyavirus, is a hantavirus.

Response. We consider that this Table is useful and to be more accurate, we have changed its legend:  “Examples of members of the Bunyavirales and associated diseases that could potentially be treated with ASA, assuming that studies with animal models and clinical trials will give good results, and those that could not be treated with ASA because they produce hemorrhagic fever. Distribution, reservoirs and vectors are included. N.I., not identified.  Sources: CDC, NIH and ECDC”.

Reviewer 2 Report

The authors explored among clinically tested compounds for antiviral agents against Bunyavirales. First, they selected five candidate inhibitors of the RNA polymerase endonuclease domain of hantavirus in silico. They then used Bunyamwera virus (BUNV) to determine whether each drug showed an inhibitory effect. One of them, Acetylsalicylic acid (ASA), was found to inhibit BUNV. However, this study does not include in vivo experimental results. In addition, the anticoagulant properties of ASA preclude its use in the major fatal viral diseases by Bunyavirales as the authors stated. The mechanism of function of ASA on BUNV is only speculative and has not actually been experimentally demonstrated. For these reasons, this paper is not of high scientific impact.

Other comments

1.       It is complicated why lines 246-247 state that ASA 15mM has no effect on cell viability, even though the CC50 is 11.24mM.

2.       The authors provide N/Gc ratios in Fig 5. However, its meaning of them is not well explained. The significance of these data should be explained in detail.

3.       What is the reason for not using RVFV, CCHFV, LFV or viruses more closely related to them that cause particularly serious diseases in humans among Bunyavirales.

Minor comments

1.       Fig3 has no single asterisk or ns, but fig3 legend has those descriptions. It is unnecessary.

2.       There appears to be a large difference between the ASA 10mM and 15mM samples in the N bands of WB in Fig 4. At a glance, it seems that there is a difference of more than 10-fold. However, there is no big difference between them in the values on the right panel.

Author Response

Reviewer 2

The authors explored among clinically tested compounds for antiviral agents against Bunyavirales. First, they selected five candidate inhibitors of the RNA polymerase endonuclease domain of hantavirus in silico. They then used Bunyamwera virus (BUNV) to determine whether each drug showed an inhibitory effect. One of them, Acetylsalicylic acid (ASA), was found to inhibit BUNV. However, this study does not include in vivo experimental results. In addition, the anticoagulant properties of ASA preclude its use in the major fatal viral diseases by Bunyavirales as the authors stated. The mechanism of function of ASA on BUNV is only speculative and has not actually been experimentally demonstrated. For these reasons, this paper is not of high scientific impact.

Response. The study does not include in vivo experiments but provides the necessary results with cell cultures for a first selection of potentially interesting antivirals. It also includes data obtained by light and electron microscopy that provide additional details. The Bunyavirales include many viruses, more than 500. ASA could be a candidate to treat infections produced by many of them although infections by those viruses producing hemorrhagic fever could not be treated with ASA. About the mechanism of action, considering that ASA was selected in silico as a potential inhibitor of the vRNApol, we believe that an action on this protein is possible, but of course we cannot exclude additional targets. This will be part of our future studies that will also investigate why the RO of the virus is not assembled in the presence of ASA.

Other comments

  1. It is complicated why lines 246-247 state that ASA 15mM has no effect on cell viability, even though the CC50 is 11.24mM.

Response: The MTT toxicity assay measures the mitochondrial dehydrogenase activity in living cells and with this test we calculated a CC50 of 11.24 mM for ASA in Vero cells. However, with 15 mM ASA, cell monolayers looked healthy by both light and electron microscopy and we decided to add this concentration for further studies.  To avoid confusion, we have suppressed the comment “concentrations that do not compromise cell viability”. 

  1. The authors provide N/Gc ratios in Fig 5. However, its meaning of them is not well explained. The significance of these data should be explained in detail.

         Response:  There was a mistake in Figure 5G, we apologize for this. What we show not the ratios but the average of the number of cells with both N and Gc protein fluorescent signals. The results show a progressive decrease of both N and Gc-associated signals with increasing concentrations of ASA, consistent with the results shown in Figure 4.  This is now corrected in Figure 5, its legend and Results.

  1. What is the reason for not using RVFV, CCHFV, LFV or viruses more closely related to them that cause particularly serious diseases in humans among Bunyavirales.

Response:  We totally agree. However, working with RVFV, CCHFV or LFV requires access to BSL3 or BSL4 labs. Positive results with inhibitors of viruses that can be handled in BSL2 labs are important before testing them with BSL3 and BSL4 pathogens, something that we plan to do in collaboration with other laboratories. We also would like to identify antivirals for a variety of members of the Bunyavirales.

Minor comments

  1. Fig3 has no single asterisk or ns, but fig3 legend has those descriptions. It is unnecessary.

Response. That is correct, single asterisk and ns are now suppressed.

  1. There appears to be a large difference between the ASA 10mM and 15mM samples in the N bands of WB in Fig 4. At a glance, it seems that there is a difference of more than 10-fold. However, there is no big difference between them in the values on the right panel.

Response. Numbers in the Table were calculated from three independent experiments and a representative image of one of them is shown.

Round 2

Reviewer 1 Report

The authors have improved the manuscript although there is correction very important that should be made in order to accept the manuscript.

As it was previously suggested, due to lack of functional assays to back up the tittle, this should be changed. The authors cannot make a tittle with descriptive presumptions but with functional assays. “There are no experiments to show that the Golgi altered morphology is functionally affecting viral particle production. Therefore, the tittle should be edited”

Author Response

The title has been changed. The new one is:

Antiviral activity of acetylsalicylic acid against Bunyamwera virus in cell culture

Reviewer 2 Report

The authors have not addressed my major comment of concern. If the in vivo data cannot be added, might I suggest that the title of this study be added to state that it is an in vitro study? 

I understand that other High BSL viruses cannot be handled. However, other readers may wonder. Why was BUNV used in this study? An additional explanation is necessary on this point.

Author Response

The authors have not addressed my major comment of concern. If the in vivo data cannot be added, might I suggest that the title of this study be added to state that it is an in vitro study? 

Response. The title has been changed. The new one is:

Antiviral activity of acetylsalicylic acid against Bunyamwera virus in cell culture

I understand that other High BSL viruses cannot be handled. However, other readers may wonder. Why was BUNV used in this study? An additional explanation is necessary on this point.

Response:

In abstract: “…5 compounds were selected and their antiviral properties studied with Bunyamwera virus (BUNV), a well characterized member of the Bunyavirales.”

Changed to: “… 5 compounds were selected and their antiviral properties studied with Bunyamwera virus (BUNV), a prototypic bunyavirus widely used for studies about the biology of this group of viruses and to test antivirals”.